# Vaginal *Atopobium* is Associated with Spontaneous Abortion in the First Trimester: a Prospective Cohort Study in China

Si Chen,[a,b] Xiaomeng Xue,[a,b] Yingxuan Zhang,[a,b] Huimin Zhang,[a,b] Xuge Huang,[a,b] Xiaofeng Chen,[a,b] Gaopi Deng,[c] Songping Luo,[c] Jie Gao[c]

[a]The First Clinical College, Guangzhou University of Chinese Medicine, Guangzhou, Guangdong, China
[b]Lingnan Medical Research Center of Guangzhou University of Chinese Medicine, Guangzhou, Guangdong, China
[c]Department of Gynecology, The First Affiliated Hospital of Guangzhou University of Chinese, Guangzhou, Guangdong China

Si Chen, Xiaomeng Xue, and Yingxuan Zhang contributed equally to this article. Author order was determined by the corresponding author after negotiation.

**ABSTRACT** Spontaneous abortion (SA) has received more and more attention in light of its increasing incidence. However, the causes and pathogenesis of SA remain largely unknown, especially for those without any pathological features. In this study, we characterized the vaginal microbiota diversity and composition of pregnant women in their first trimester and evaluated the association between the vaginal microbiota and SA before 12 weeks of gestation. Participants' bacterial profiles were analyzed by 16S rRNA gene sequencing in the V3–V4 regions at 5–8 weeks of gestation. A total of 48 patients with SA at 12 weeks of gestation were included as the study group, while 116 women with normal pregnancies (NPs) were included as a control group. The results indicated that the richness of the vaginal microbiome in SA patients was higher (Chao1, $P < 0.05$) and different in composition relative to that of women with NPs (unweighted UniFrac, $R = 0.15$, $P < 0.01$; binary Jaccard, $R = 0.15$, $P < 0.01$). Furthermore, the genus *Apotobium* was significantly enriched in SA patients. An extreme gradient-boosting (XGBoost) analysis was able to classify *Atopobium*-induced SA more reliably (area under the receiver operating characteristic curve, 0.69; threshold, 0.01%). Moreover, after adjusting for potential confounders, the results showed a robust association between *Apotobium* and SA (as a categorical variable [<0.01%]; adjusted odds ratio, 2.9; 95% confidence interval, 1.3 to 6.5; $P = 0.01$). In conclusion, higher vaginal *Apotobium* levels were associated with SA in the first trimester.

**IMPORTANCE** Spontaneous abortion (SA) is the most common adverse pregnancy outcome in the first trimester. The causal drivers of SA have become a substantial challenge to reveal and overcome. We hypothesize that vaginal microbial dysbiosis is associated with SA, as it was related to several female reproductive disorders in previous studies. In our study, we characterized the vaginal microbiota of patients with SA at 12 weeks of gestation as the study group, and women with normal pregnancies were enrolled as a control group. Generally, significant differences were discovered in the vaginal microbiota between the two groups. Our study also revealed that *Apotobium* may play an important role in the pathogenesis of SA. To our knowledge, this study is the first detailed elaboration of the vaginal microbiota composition and vaginal *Apotobium* in association with SA. We believe that our findings will inspire more researchers to consider dynamic changes in the vaginal microbiota as critical features for further studies of nosogenesis not only for SA but also other reproductive diseases.

**KEYWORDS** spontaneous abortion, vaginal microbiota, 16S rRNA, machine learning, *Atopobium*

Address correspondence to Jie Gao, gaojie1769@gzucm.edu.cn, Songping Luo, songpingluo@hotmail.com, or Gaopi Deng, denggaopi@126.com.

The authors declare no conflict of interest.

Spontaneous abortion (SA) is defined as a loss of pregnancy before the fetus reaches viability. As one of the most common complications to occur in early pregnancy, the incidence of SA has reached 15.3% of all recognized pregnancies (1). Moreover, >80% of SA cases occur in the first 12 weeks of gestation (2). SA has therefore become a definite global reproductive health issue, leading to adverse effects on physical and mental health and increasing economic pressure (3, 4). The management of SA usually centers on the blood or urinary $\beta$-subunit of human chorionic gonadotropin ($\beta$-hCG) test results. However, although pregnant women may undergo several $\beta$-hCG tests and attempt various possible management techniques, some will still inevitably experience SA before effective treatment is completed. Moreover, the exact etiology of SA has not been well identified thus far. Reported possible maternal factors include chromosomal abnormalities (1, 5), endocrinological causes (6), immune system influence (6, 7), maternal age (8), environmental factors (1), and an unhealthy lifestyle (9–11). Meanwhile, some pregnant women with SA have none of the pathological features mentioned above; therefore, it is necessary to identify additional risk factors, and there is an urgent need to find multidimensional biomarkers to assist with the early prediction of SA.

The microbiota plays a fundamental role in the overall development and defense of human beings. The vagina is a unique and complex ecosystem, and a balance between the vaginal microbial communities is vital for female health (12). Vaginal microbiota dysbiosis in women of reproductive age is associated with plenty of diseases, particularly reproductive disorders, including preterm delivery (13), intrauterine adhesions (14), and infertility (15). Fortunately, knowledge has been rapidly accumulating owing to the increased sensitivity of microbial detection methods like 16S rRNA gene sequencing (16). However, few studies to date have focused on the association between the vaginal microbiota and SA. Because of the diversity of the vaginal microbiota among different ethnicities and given the huge population base in China, there is an urgent and unavoidable need to explore the association between the vaginal microbiota and SA in China.

In our study, we performed a prospective cohort investigation and aimed to assess the underlying association between the vaginal microbial composition and the incidence of SA among pregnant women in their first trimester in China. In order to perform a better analysis, both 16S rRNA gene sequencing and machine learning were used as advanced analysis tools. We tried to provide a new perspective on a potential microbial biomarker for SA.

## RESULTS

**Patient selection.** Among 268 women with positive pregnancy tests (Fig. S1 in the supplemental material), 80 pregnant women diagnosed with ectopic pregnancy were excluded from this study. We continued to follow up with the remaining 188 participants until 12 weeks of gestation to confirm the pregnancy status. Subsequently, we excluded 24 participants due to missing follow-up data. Finally, 164 participants were included and eligible for 16S rRNA sequencing. Among them, 116 were in the normal pregnancy (NP) group and 48 were in the SA group.

**Baseline characteristics of the study population.** The baseline characteristics are listed in Table 1. Compared to those in the NP group, participants in the SA group were more likely to have a higher relative abundance of *Apotobium* ($P < 0.01$). Furthermore, participants in the SA group had significantly more gestational days ($P < 0.05$). There was no statistically significant difference in other factors, including age, body mass index, smoking history, symptoms, menstrual cycle, previous pregnancy history, previous surgery history, previous pelvic inflammatory disease, baseline $\beta$-HCG, progestin, estradiol, and vaginal pH.

**Vaginal microbial characteristics of NP and SA women.** Vaginal microbiota samples from 116 NP and 48 SA women were analyzed by 16S rRNA sequencing. With respect to $\alpha$-diversity, compared to NP participants, participants in the SA group showed higher Chao1, observed species, and phylogenetic diversity whole tree index

**TABLE 1** Baseline characteristics of participants

| Diagnosis | NP[a] | SA | P-value[b] |
|---|---|---|---|
| No. | 116 | 48 | |
| BMI[c] | 28.3 ± 6.1 | 27.2 ± 7.4 | 0.34 |
| Age, yr | 29.3 ± 5.5 | 30.8 ± 5.5 | 0.13 |
| Current or ex-smoker, yes | 0.0 ± 0.0 | 0.0 ± 0.0 | / |
| Gestational age, days | 47.3 ± 7.2 | 50.2 ± 9.1 | 0.03* |
| Uterine bleeding, yes | 42 (41.2%) | 22 (48.9%) | 0.39 |
| Abdominal pain, yes | 30 (27.8%) | 17 (36.2%) | 0.30 |
| Menstrual cycle, days | 38.7 ± 16.0 | 34.6 ± 9.9 | 0.18 |
| Gravidity | 2.6 ± 1.5 | 2.2 ± 1.1 | 0.17 |
| Previous vaginal delivery, yes | 38 (36.2%) | 10 (21.3%) | 0.07 |
| Previous cesarean delivery, yes | 15 (14.3%) | 6 (12.8%) | 0.80 |
| Previous ectopic pregnancy, yes | 3 (2.9%) | 1 (2.1%) | 0.80 |
| Previous pregnancy loss, yes | 51 (44.3%) | 14 (29.2%) | 0.07 |
| Previous uterine cavity surgery, yes | 34 (55.7%) | 16 (45.7%) | 0.34 |
| Previous pelvic surgery, yes | 19 (16.5%) | 9 (18.8%) | 0.73 |
| Previous pelvic inflammatory disease, yes | 16 (13.8%) | 6 (12.5%) | 0.82 |
| Baseline HCG log10 transform | 4.1 ± 0.9 | 3.9 ± 1.0 | 0.42 |
| Baseline progestin | 20.8 ± 13.8 | 18.5 ± 11.3 | 0.37 |
| Baseline estradiol | 1035.1 ± 1584.9 | 1450.6 ± 1850.0 | 0.23 |
| Vaginal environments | | | |
| Vaginal PH | 4.0 ± 1.6 | 4.2 ± 1.2 | 0.43 |
| *Atopobium* group | | | <0.01*[d] |
| <0.01% | 71 (61.2%) | 17 (35.4%) | |
| ≥0.01% | 45 (38.8%) | 31 (64.6%) | |

[a]Continuous variables were presented as mean ± SD, categorical variables were expressed as percentages (%).
[b]P was calculated by *t* test for normally distributed continuous variables, chi-squared test or Fisher's exact test for categorical variables.
[c]BMI: body mass index.
[d]*, P < 0.05.

values ($P < 0.05$), but there was no significant difference in the Shannon index between the two groups. This means that the community richness of the vaginal microbiota in the SA group was greater than that in the NP group, while there was no significant difference in community diversity between the two groups (Fig. S2 in the supplemental material). Next, we undertook PCoA to further evaluate the $\beta$-diversity in the NP and SA groups (Fig. S3). An analysis of similarities indicated that the dissimilarity between two groups was more significant than that within the groups (binary Jaccard, $R = 0.15$, $P < 0.01$; unweighted UniFrac, $R = 0.15$, $P < 0.01$). Taking the phylogeny and abundance into consideration, the vaginal microbial composition of pregnant women in the NP group was different than that of pregnant women in the SA group.

For the purpose of visualization, we presented the top 15 taxa at the phylum and genus levels, respectively, and genera listed after the top 15 taxa were incorporated together as "others." In Fig. 1, the composition of the vaginal microbiota in the NP and SA groups was similar, and the relative abundances of different vaginal microbial species were slightly different. The vaginal microbiota of the NP and SA groups were dominated by *Firmicutes* (NP, 85.7%; SA, 85.2%), *Actinobacteria* (NP, 9.0%; SA, 9.9%), and *Bacteroidetes* (NP, 3.3%; SA, 3.1%). At the genus level, the composition between the two groups was similar, but the relative abundance of some species was significantly different; *Lactobacillus* (NP, 81.2%; SA, 79.3%), *Gardnerella* (NP, 5.6%; SA, 5.7%), *Atopobium* (NP, 2.0%; SA, 3.5%), *Prevotella* (NP, 2.5%; SA, 1.2%), and *Streptococcus* (NP, 1.4%; SA, 1.7%) were the main components.

Next, to further explore the potential taxonomic biomarkers characterizing the differences between NP and SA participants, LEfSe analysis with a logarithmic LDA value of 2.0 was performed (Fig. 2). *Corobacteriia*, *Coriobacteriales*, *Atopobiaceae*, *Atopobium*, *Streptococcaceae*, and other 4 kinds of taxa were enriched in the SA group (Fig. 2A). At the genus level, *Atopobium* was deemed worthy of attention in the SA group. Moreover, for further analysis, we chose the taxa screened by LEfSe analysis and

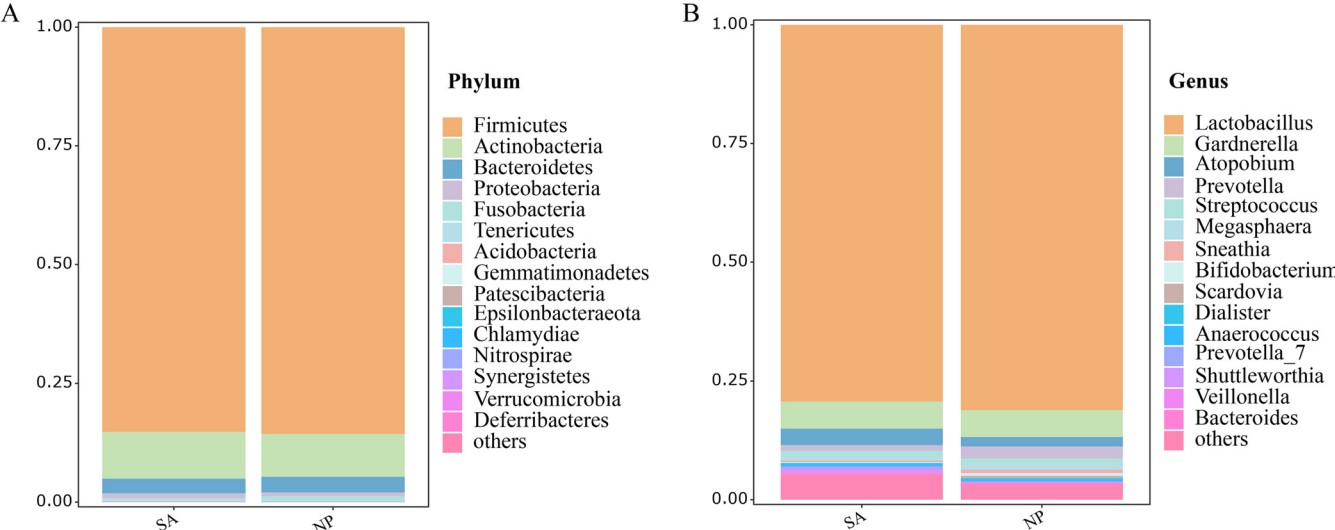

**FIG 1** Vaginal taxonomic profiles of participants in the normal pregnancy and spontaneous abortion groups. (A) Phylum level. (B) Genus level. Bar charts showing the vaginal microbial taxa composition in mean values.

compared the top 15 relative abundances in the NP and SA groups (Fig. 2B). The relative abundances of *Atopobium*, *Prevotella_7*, *Shuttleworthia*, and *Bacteroides* in the SA group were significantly greater than those in the NP group ($P < 0.05$), while the relative abundances of *Sneathia* and *Bifidobacterium* in the SA group were significantly smaller than those in the NP group ($P < 0.05$). These findings suggested to us that *Atopobium* at the genus level might play an essential role in the classification of SA.

**Atopobium shows a potential microbial predictive value with early pregnancy outcome.** Based on the vaginal microbial features, several bacterial at the genus level were likely to be potential taxonomic biomarkers for early pregnancy outcome. To further clarify the underlying association, extreme gradient boosting (XGBoost) analysis was performed to assess the predictive value (Fig. 3A), and the results indicated that *Atopobium*, *Prevotella*, and *Escherichia.shigella* were the top 3 most important taxa associated with SA (in order of relative importance). Moreover, we assessed the predictive value of *Atopobium*, *Prevotella*, and *Escherichia.shigella* by ROC curve analysis. The average ROC curve values of *Atopobium*, *Prevotella*, and *Escherichia.shigella* were 0.686, 0.515, and 0.652, respectively (Fig. 3B). Considered in combination with LEfSe analysis, the relative abundance of *Atopobium* was the most essential microbial biomarker associated with SA in early pregnancy, and the best threshold was 0.01% (Table S1 in the supplemental material). We used 0.01% as the cutoff value and defined the relative abundance of *Atopobium* as a categorical variable for subsequent analysis. As shown in Table 1, the percentage of participants in the SA group with a relative abundance of *Atopobium* >0.01% was greater than that in the NP group (NP, 38.8%; SA, 64.4%; $P < 0.01$).

**The relative abundance of vaginal Atopobium was stably associated with SA.** We further evaluated the association between the relative abundance of *Atopobium* and SA by multivariate logistic regression. The association between *Atopobium* and SA was robust in the crude, minimally adjusted, and fully adjusted models (Table 2). The *Atopobium* group ($\geq$0.01%) showed a robust and positive association with SA (OR, 2.9; 95% confidence interval [CI], 1.4 to 5.8; $P < 0.01$ [crude model]; OR, 2.3; 95% CI, 1.1 to 4.9; $P = 0.03$ [minimally adjusted model]; OR, 2.9; 95% CI, 1.3 to 6.5; $P = 0.01$ [fully adjusted model]). Therefore, we could naturally reach the conclusion that the relative abundance of vaginal *Atopobium* in early pregnant women is stably associated with the incidence of SA.

## DISCUSSION

**Main findings.** Both a greater richness and abundance of *Atopobium* in the vaginal microbiota were associated with SA during 5–8 weeks of gestation. As a potential

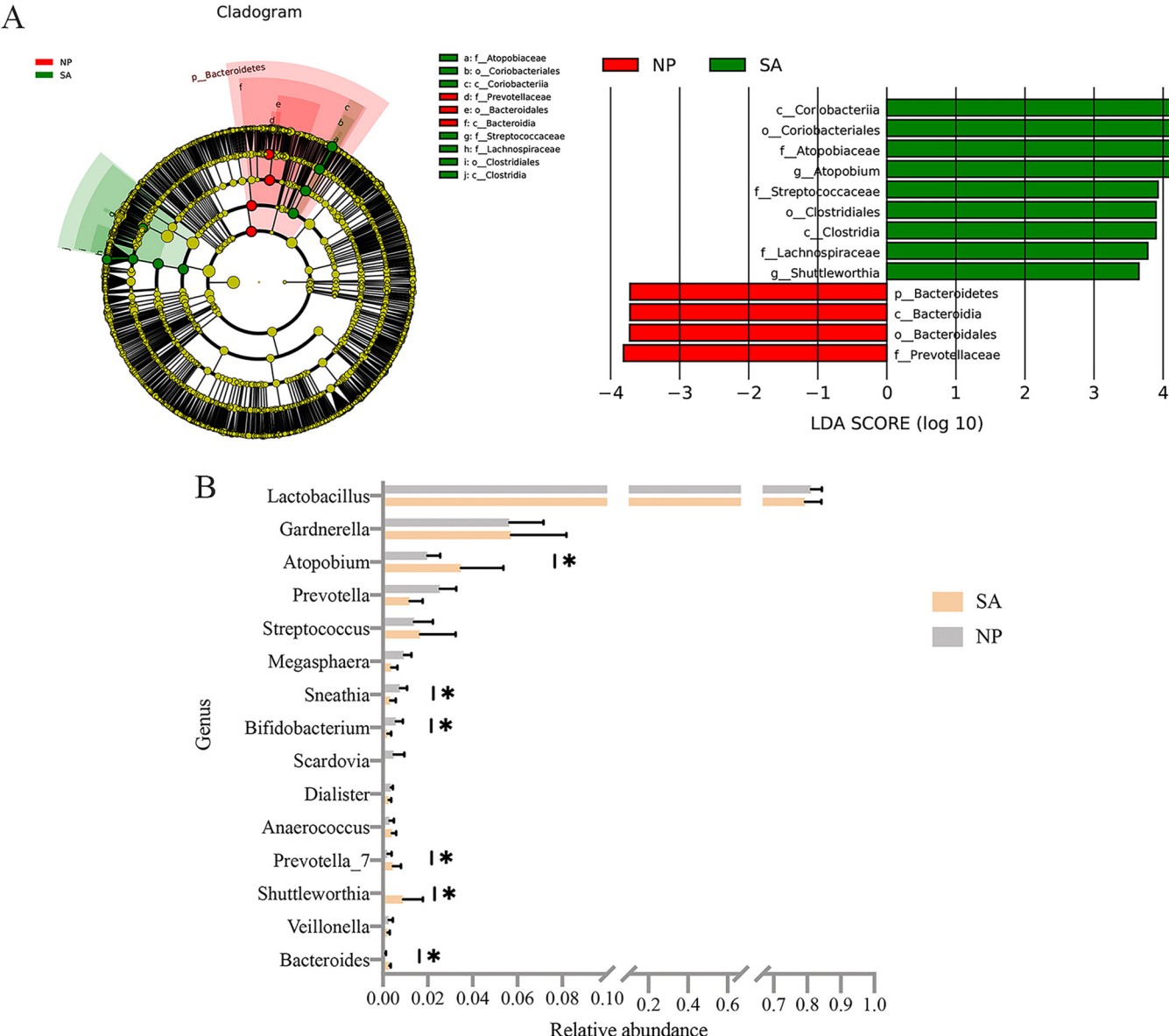

**FIG 2** Different vaginal microbiota compositions of the normal pregnancy (NP) and spontaneous abortion (SA) groups. (A) Cladogram analyzed by linear discriminant analysis (LDA) by LEfSe. The nodes of the cladogram from the inner to the outer circles showed the abundant taxa from the kingdom (D0) level to the species (D6) level. Colors represented the groups in which differentially abundant taxa were enriched (red indicates NP, green indicates SA, yellow indicates non-significant), and the diameter of each node was proportional to the taxon's abundance. The threshold on the logarithmic LDA score was 2.0. (B) The levels of relative abundance of *Lactobacillus*, *Gardenerella*, *Atopobium*, *Prevotella*, *Streptococcus*, *Sneathia*, *Bifidobacterium*, etc. were compared in individuals in the NP and SA groups. The *P* value was determined by the two-tailed Wilcoxon rank-sum test.

microbial biomarker, the relative abundance of *Atopobium* could be used to predict SA in the first trimester by the threshold of 0.01%. When the relative abundance of *Atopobium* is >0.01% at 5–8 weeks of gestation, it is positively associated with the incidence of SA in the first trimester.

**Interpretation.** The development of sequencing technologies and analysis tools has promoted exploration and understanding of the human microbiota (17). Recently, growing evidence has come to support the idea that abnormal alterations in the vaginal microbiota are related to several female reproductive disorders (18, 19). Although epidemiological research showed that women with disorders of the vaginal microbiota rarely manifest symptoms of infection (20), the unremarkable and common microorganisms in play in such an environment remain dangerous and can lead to preterm delivery (13, 21), failure of *in vitro* fertilization (22, 23), infertility (18), and so on.

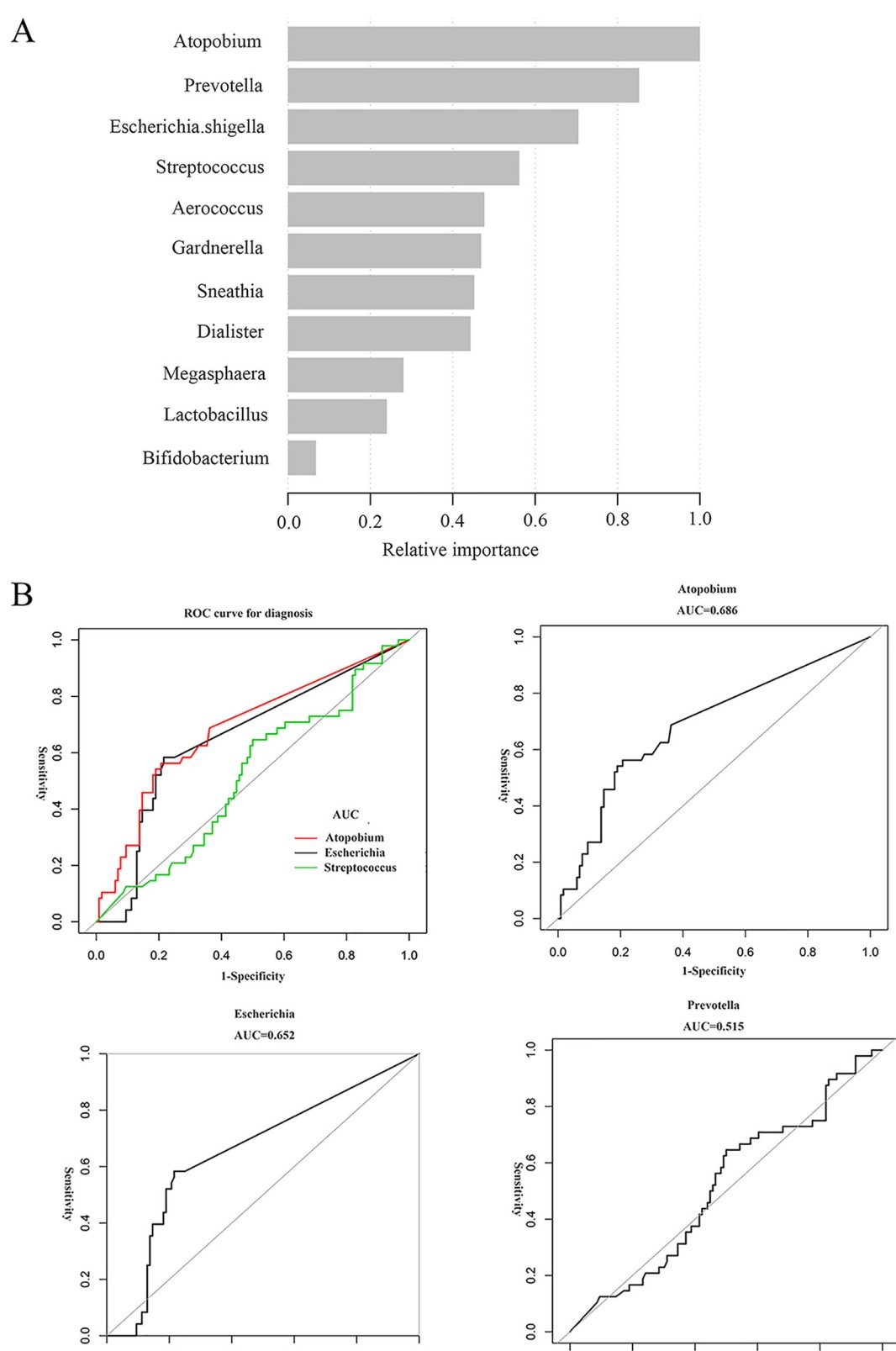

**FIG 3** Screening the potential importance of the top 15 taxa at the genus level on the early pregnancy outcome by XGBoost. (A) XGBoost analysis was applied to assess the weight of different taxa at the genus level on the early pregnancy outcome. The analysis showed that the genus *Atopobium* was the one most related to SA. (B) Receiver operating characteristic (ROC) curve analysis was employed to evaluate the predictive value of *Atopobium*, *Esherichia*, and *Prevotella*. The black curve shows the average area under

**TABLE 2** Multivariate logistic regression analysis for the relationship between relative abundance of *Atopobium* and SA

| Abundance range | Crude model[a] OR (95% CI) *P* | Adjusted I model[b] OR (95% CI) *P* | Adjusted II model[c] OR (95% CI) *P* |
|---|---|---|---|
| *Atopobium* group | | | |
| <0.01% | 1.0 | 1.0 | 1.0 |
| ≥0.01% | 2.9 (1.4, 5.8) <0.01 | 2.3 (1.1, 4.9) 0.03 | 2.8 (1.3, 6.3) 0.01 |

[a]Crude model: adjusted for: none.
[b]Adjusted I Model: adjusted for: age, body mass index (BMI).
[c]Adjusted II Model: adjusted for: age; BMI; gestational age; uterine bleeding; previous vaginal delivery; previous pregnancy loss; previous pelvic surgery; previous ectopic pregnancy; previous pelvic inflammatory disease.

As the most common adverse pregnancy outcome, SA affects 20–25% of pregnancies (24, 25). Approximately 50% of all SA cases are considered to be caused by aneuploidy or other chromosomal abnormalities, while the casual etiology of the remaining cases has yet to be fully elucidated (25). Increasing evidence supports the idea that inflammation of the female reproductive tract increases the incidence of SA (6). Recent research has also indicated that, compared to samples from a control group (0%), histological chorioamnionitis was detected in most samples of the SA group (77%); also, among them, 47% of chorioamnionitis cases were culture-positive (26). Moreover, other studies have found an association between bacterial vaginosis (BV) and the risk of SA (27–29).

Members of the genus *Atopobium* are commonly found in the vaginas of women, with 97% of these being *A. vaginae* (30), a Gram-positive, strict anaerobe that produces major amounts of lactic acid (31). This kind of bacteria has been considered a common component of the vaginal microbiome, but a series of studies found that *Atopobium* in the female genital tract is associated with different female reproductive disorders; for example, an *A. vaginae*–dominated vagitype was observed in pregnant women who delivered preterm babies in a Nigerian cohort (32), and women at high risk of preterm birth were identifiable by a high vaginal load of *A. vaginae* (DNA level ≥ 108 copies/mL) (33). Second, previous study has suggested *Atopobium* to be one of the significantly more abundant genera associated with recurrent spontaneous abortion (34). Third, a study proposed that *Atopobium* spp. is a microbial marker of human papillomavirus persistence in the cervico-vaginal microbiome (35). Fourth, a greater abundance of *Atopobium* has also been linked to more severe pelvic inflammatory disease, endometritis, and infertility (36, 37). Besides, *Atopobium* spp. is an important component of the disturbed vaginal microbiome of women with BV, and *A. vaginae* can be detected by PCR in 96% of those with BV (38). *A. vaginae* has been associated with 75% of the Amsel clinical criteria, including vaginal discharge, elevated pH, and the presence of clue cells (39). Although the BV etiology remains unclear, one hypothesis suggested that virulent strains initiate the formation of biofilm on vaginal epithelial cells and become a scaffolding to which other BV-associated anaerobes thereafter can attach. *A. vaginae* is one of the species associated with virulent strains of *Gardnerella* spp. that dominate the biofilm on the vaginal epithelium (40–42). Moreover, a relationship between BV and an increased incidence of SA has been reported (27–29, 43). Endometritis is thought to lead to SA through the impairment of endometrial decidualization (44, 45). Therefore, we hypothesized that the association between *Apotobium* and SA might be related to BV. The underlying mechanism may be as follows: *A. vaginae* has a unique potential to evoke an immune response and disrupts physicochemical barrier properties of the vaginal mucosa by inducing a broad range of proinflammatory cytokines, chemokines, and antimicrobial peptides, including interleukin (IL)-1 $\beta$,

**FIG 3 Legend (Continued)**

the ROC curve (AUC) of these taxa. The diagonal lines mark an AUC of 0.5. (Upper left) Summary ROC figure. Red indicates *Atopobium*, gray indicates *Escherichia*, and green indicates *Prevotella*. (Upper right) ROC of *Atopobium* (AUC, 0.686). (Lower left) ROC of *Escherichia* (AUC, 0.652). (Lower right) ROC of *Prevotella* (AUC, 0.515).

IL-6, IL-8, macrophage inflammatory protein (MIP)-3$\alpha$, tumor necrosis factor, and human $\beta$-defensin 2 (46, 47). Other reports have demonstrated that *A. vaginae* induces human $\beta$-defensin 4, MIP-1$\beta$, growth-regulated oncogene $\alpha$, and granulocyte colony-stimulating factor, and the host immune response was Toll-like receptor (TLR) 2–dependent (48–50). Higher concentrations of *A. vaginae* are associated with significantly lower concentrations of human $\beta$-defensin 3 (51). Besides, it has been shown that some detrimental vaginal microorganisms can enter the uterine cavity by ascending from the vagina and cervix, causing intrauterine infections, which may also explain the association between *Atopobium* and SA (52, 53).

On the other hand, it remains a great challenge to translate complex ecological metrics into the clinical setting. Classifying the vaginal microbiota effectively led to systematic listing of the complicated biological data set, and it is also beneficial for epidemiological research and disease diagnosis (36, 54). As in previous reports, community state type (CST) classification was performed to cluster the vaginal microbiota (55). Four of the CSTs are dominated by *Lactobacillus*; meanwhile, only CST IV was dominated by several anaerobic bacteria, including *Atopoium*, *Gardnerella*, *Prevotella*, and *Mobiluncus*. Actually, about 25% of the women sampled were clustered into CST IV (56). Therefore, the drawbacks of CST classification lead to a lack of a concentrated detrimental effect of *Atopobium*. Recently, several studies have proposed that classifications based on relative abundance at the genus level could classify samples more clearly and assess the association between the vaginal microbiota and pregnancy outcome more precisely (57). Robinson et al. proposed that machine learning algorithms could be applied to overcome such defects (58). Machine learning algorithms could effectively promote bioinformatics analysis for making the microbial-community groups independent of samples and assisting with comparability between studies (59). As an efficient algorithm of machine learning, XGBoost has been widely applied to predict the onset of disease (60, 61). The predictive testing process of the XGBoost model is conducive to the assessment of expecting the onset of disease, and it helps to optimize medical measurements (62). Therefore, we combined LEfSe analysis and the XGBoost algorithm to effectively process the large-scale microbial data to achieve the identification of SA.

**Strengths and limitations.** Our study has several implications. First, we found an underlying link between abnormal alterations in the vaginal microbiota and SA. We also provided a new insight that the relative abundance of *Atopobium* is a potential microbial biomarker for the incidence of SA. Further, this study provides data support for future clinical trials. In addition, only limited studies to date have focused on the association between vaginal microbiota and SA in the first trimester, making our study somewhat unique. Finally, we combined LEfSe analysis and a machine learning algorithm to classify the incidence of SA at 5–8 weeks of gestation more reliably.

Although our study tried to provide comprehensive insight into the relationship between the vaginal microbiota and SA, some limitations remain to be addressed. First, because the composition of the vaginal microbiota may be influenced by ethnicity, further studies involving larger sample sizes of populations outside of China are necessary. Second, our focus in this study was on vaginal bacteria in women with early pregnancy (12 weeks of gestation), and future studies could extend the follow-up to the mid- or late pregnancy period. Third, due to the limitations of 16S rRNA sequencing, the discussion of fungi was not addressed in this study, and we believed that fungi should also be included in future research. Additionally, relevant clinicopathological data with larger sample size were also meaningful to verify and strengthen the findings of this study. Finally, although we found that the relative abundance of *Atopobium* could be a potential and valuable biomarker associated with SA, the underlying mechanism needs to be verified by further animal experiments.

## CONCLUSION

We showed that both a greater richness and abundance of *Atopobium* in the

vaginal microbiota were associated with the incidence of SA. The relative abundance of *Atopobium* (threshold ≥ 0.01%) in the vaginal microbiota could be a predictive biomarker.

## MATERIALS AND METHODS

**Ethical approval.** This study was approved by the ethics committee of the First Affiliated Hospital of Guangzhou University of Chinese Medicine (no. ZYYECK2017-060) located in Guangzhou, China. All participants included in this study signed an informed consent form, and all processes in our study followed the tenets of the Declaration of Helsinki.

**Study design.** This prospective cohort study was a secondary analysis of a follow-up cohort of pregnant women in China. From May 2018 to December 2020, women with a positive pregnancy test (N = 268) and between 5–8 weeks of gestation initially were enrolled in this study at the First Affiliated Hospital of Guangzhou University of Chinese Medicine.

The inclusion criteria were as follows: (1) pregnancy test result was positive; (2) age > 18 years; and (3) between 5 and 8 gestational weeks (from the last day of the last menstruation) at enrollment. The exclusion criteria were as follows: (1) ectopic pregnancy; (2) lost to follow-up; (3) took antibiotics within 30 days before sample collection; (4) had sexual activity, performed vaginal douching, or recorded the use of vaginal drugs within 48 h before sample collection; and (5) presented with acute inflammation, cancer, or vulvovaginal candidiasis.

Participants in this study were followed up with until 12 weeks of gestation. The pregnancy outcome was identified by both clinical symptoms, physical examination, repeated female endocrine testing, and vaginal ultrasonography. Cases in this study were defined as women diagnosed with SA, including those who experienced complete, incomplete, or missed abortion, before 12 weeks of gestation. Control group members were pregnant women attending antenatal care before 12 weeks of gestation who did not have a tendency for SA (24, 63–66).

**Collection of clinical data and samples.** When a pregnant woman visited our hospital initially, she was asked to fill out a case report form, which requested details on sociodemographic characteristics, past medical/reproductive history, lifestyle, symptoms, and measurements of height and weight. Each patient's baseline serum $\beta$-HCG concentration, progesterone level, and ultrasound findings were recorded. The collection of vaginal secretion samples was also completed by an experienced gynecologist after obtaining patient consent at baseline. Three sterile swabs (Improve Medical, Guangzhou, China) as triplicates were applied to each participant's samples. All samples were collected from both sides of the mid-vaginal canal after a sterile speculum without lubrication was inserted into the vaginal canal, 5 times in total. Swabs were placed individually in sterile tubes. One swab was used for screening and for excluding vulvovaginal candidiasis by wet mount microscopy (67). One swab was sent immediately to measure the pH of the vaginal secretions. One swab was transported to the laboratory and stored at −80°C until DNA extraction.

**Determination of vaginal pH.** Vaginal pH was measured using pH test paper (Sanaisi Scientific Instruments, Jiangsu, China) on an automatic vaginitis detection system (bPR-2014A; Bioperfectus Technologies, Jiangsu, China) and ranged from 3.8 to 5.4 (incremental change, 0.2).

**Total DNA extraction and 16S rRNA sequencing.** Total genomic DNA was prepared using the DNeasy PowerSoil Kit (Qiagen, Hilden, Germany). Agarose gel was used to verify the concentration of DNA. The eligibility criteria for library construction and sequencing included 1) an obvious DNA main band, 2) DNA concentration > 10 ng/$\mu$L, and 3) an OD$_{260}$/OD$_{280}$ ratio > 1. We used AMPure XP beads (Agencourt Bioscience Corporation, Beverly, MA, USA) for purification and performed another round of PCR amplification for qualified samples. After purification with AMPure XP magnetic beads again, the final amplicon was quantified using the Qubit dsDNA Detection Kit (Invitrogen, Carlsbad, CA, USA). We proceeded to the next step of sequencing after merging equal amounts of purified amplicons. The amplified region was the corresponding region for bacterial diversity identification: 16S V3–V4 region (upstream primer 343F, 5'TACGGRAGGCAGCAG-3; downstream primer 798R, 5-AGGGTATCTAATCCT3'). The samples were sequenced using the MiSeq platform (Illumina, San Diego, CA, USA), and the sequencing strategy was PE300.

Sequencing raw data were stored in FASTQ format. The Trimmomatic version 0.35 software tool was used to perform primer, label removal, and quality control (67). We used FLASH (version 1.2.11) to splice double-ended sequences with an overlap of 10–200 bp and a mismatch rate of <20% to establish a total paired-end sequence (68). The ambiguous N bases and sequences with single base replicates >8 and lengths <200 bp were divided by Split_libraries (version 1.830) to obtain clean tags (69). After removing the chimera by UCHIME (version 2.4.2) (70), the valid tags separated by operational taxonomic units (OTUs) were acquired for the following analyses (71). After preprocessing has been accomplished, Vsearch (version 2.3.2) was used to cluster the sequences with resemblance ≥97% and incorporate them into a single OTU (72). We picked the sequence with the highest richness as the representative sequence and compared the typical sequence against the Greengenes and SILVA databases for annotation and classification (73).

**Data analysis.** Based on the number of sequences included in OTUs, the OTU matrix file for the following analysis and calculation was established. The QIIME software was used to calculate $\alpha$-diversity and $\beta$-diversity; $\alpha$-diversity (Chao1, observed species, phylogenetic diversity whole tree, and Shannon diversity index) was calculated to assess the within-community diversity, while $\beta$-diversity (binary Jaccard and unweighted UniFrac) was calculated to evaluate the alteration in community composition (71). Principal-component analysis (PCoA) was applied for further analysis of the vaginal microbiota

diversity of the community structure. Linear discriminant analysis (LDA) effect size (LEfSe) with default parameters ($\alpha$-value for the Wilcoxon test, 0.05; threshold of the logarithmic LDA score, 2.0 points) was used to find species with significant differences between different groups (74).

To evaluate the importance of the relative abundance, we employed XGBoost, a machine learning method. The accuracy of species was compared by the average area under the receiver operating characteristic (ROC) curve (AUC). Moreover, a multivariate logistic regression analysis was performed to evaluate the potential association between *Apotobium* and SA. Crude, minimally, and fully adjusted models were all established. Covariances added in the adjusted model all had matching odds ratios (ORs) with alterations >10%. Both XGBoost analysis and multivariate logistic regression analysis were performed using EmpowerStats version 2.2 (www.empowerstats.com/).

The baseline characteristics of the study population and the index of $\alpha$-diversity were analyzed using SPSS version 22.0 (IBM Corporation, Armonk, NY, USA). Differences between groups were assessed with the Student's *t* test for continuous variables with a normal distribution, the Wilcoxon test for skewed continuous variables, and the chi-squared test or Fisher's exact test for categorical variables, respectively. $P < 0.05$ (2-tailed) was considered to be statistically significant.

**Data availability.** All sequence data generated from this study were deposited in the NCBI Sequence Read Archive under BioProject accession PRJNA737055.

## SUPPLEMENTAL MATERIAL

Supplemental material is available online only.

**SUPPLEMENTAL FILE 1**, PDF file, 0.5 MB.

## ACKNOWLEDGMENTS

We gratefully thank all participants of the study and members of the Department of Gynecology, First Affiliated Hospital of Guangzhou University of Chinese Medicine, and the Lingnan Medical Research Center of Guangzhou University of Chinese Medicine. We also thank Xiaorong Liu and Xiaofeng Ruan for their contributions to the sample collection.

The ethics committee of the First Affiliated Hospital of Guangzhou University of Chinese Medicine (no. ZYYECK2017-060), Guangzhou, China, reviewed and approved this study.

The study was funded by the National Natural Science Foundation of China (81774358), the Guangzhou University of Chinese Medicine (XK2019016 and 2019IIT33), the National Administration of Traditional Chinese Medicine (National Chinese Medicine People's Education Development [2018] no. 12), and the Department of Finance of Guangdong Province (2020B11111100003).

We declare that no conflicts of interest exist.

J.G., S.P.L., and G.P.D. designed the study. J.G. and S.P.L. founded the study. X.F.C., X.G.H., X.M.X., and H.M.Z. recruited participants. S.C., Y.X.Z., X.M.X., and H.M.Z. collected clinical data and samples as well as analyzed and interpreted the data. S.C. and Y.X.Z. generated all figures and tables. S.C. and X.M.X. wrote the first draft, which was further developed by H.M.Z. and Y.X.Z. All authors have read and approved the final manuscript.

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
