## [Reviewer comments · Microbiology Spectrum]

Microbiology Spectrum

Vaginal *Atopobium* is associated with spontaneous abortion in first trimester: a prospective cohort study in China

Si Chen, Xiaomeng Xue, Yingxuan Zhang, Huimin Zhang, Xuge Huang, Xiaofeng Chen, Jie Gao, Songping Luo, and Gaopi Deng

Corresponding Author(s): Jie Gao, First Affiliated Hospital of Guangzhou University of Chinese Medicine

Review Timeline:

Submission Date:	October 26, 2021
Editorial Decision:	December 8, 2021
Revision Received:	January 11, 2022
Accepted:	January 14, 2022

Editor: Kevin Theis

Reviewer(s): Disclosure of reviewer identity is with reference to reviewer comments included in decision letter(s). The following individuals involved in review of your submission have agreed to reveal their identity: Zhendong Cai (Reviewer #2)

Transaction Report:

DOI: <https://doi.org/10.1128/Spectrum.02039-21>

December 8, 2021

Dr. Jie Gao
First Affiliated Hospital of Guangzhou University of Chinese Medicine
Guangzhou
China

Re: Spectrum02039-21 (Vaginal *Atopobium* is associated with spontaneous abortion in first trimester: a prospective cohort study in China)

Dear Dr. Jie Gao:

Link Not Available

Sincerely,

Kevin Theis

Journals Department
Reviewer comments:

Reviewer #1 (Public repository details (Required)):

16S sequence data should be deposited in NCBI database

Reviewer #1 (Comments for the Author):

In this study, the authors sought to characterize the vaginal microbiome composition of 164 Chinese women and describe its association with spontaneous abortion using routine microbiological method and 16S ribosomal sequencing targeting V3-V4 region.

Comments

1) Abstract should be rewritten and thoroughly revised. Authors should avoid the use of abbreviation on the abstract without defining them initially. Manuscript should be revised for typos and grammatical expression. The English in the present manuscript is not of publication quality and require major improvement. Please carefully proof-read spell check to eliminate grammatical errors.

2) In this study, vaginal PH was measured for all the women before selecting 164 vaginal swabs for 16S rRNA sequencing. However, we do not have pertinent information on the result of vaginal PH for these women. Briefly describe this in a table or figure.

3) The only exclusion criteria mentioned in the manuscript were as follows: (1) ectopic pregnancy; (2) lost follow-up; (3) took antibiotics within 30 days before sample collection; (4) had sexual activity, vaginal douching or recorded used of vaginal drugs within 48 hours before sample collection; (5) with acute inflammation, cancer of vulvovaginal candidiasis. Spontaneous Abortion can also be caused by infection. Can the authors provide information on whether the women had any history of maternal infection, history of STDs or any intercurrent disease at the course of pregnancy or previous pregnancy?

Reviewer #2 (Public repository details (Required)):

Data of 16S rRNA gene sequencing should be deposited in a public repository

Reviewer #2 (Comments for the Author):

1) I suggest that the quality of this paper as currently written is not good enough for publication. It is required for English language polishing; you can have your paper professionally edited for English language. For the first time, the full name of NP and SA should be shown in the "Abstract" section.

2) Through 16S rRNA gene sequencing, the authors demonstrated that Apotobium in the vaginal microbiota is associated with SA. However, we could not rule out the influence of pathogenic fungi such as Candida on SA, which might play important roles in the occurrence of SA.

3) The authors should supplement the clinicopathologic data such as histopathological analysis and analyze the potential correlation with vaginal microbiota, which contributes to make a comprehensive understanding.

Staff Comments:

Preparing Revision Guidelines

Please return the manuscript within 60 days; if you cannot complete the modification within this time period, please contact me. If you do not wish to modify the manuscript and prefer to submit it to another journal, please notify me of your decision immediately so that the manuscript may be formally withdrawn from consideration by Microbiology Spectrum.

Comments

In this study, the authors sought to characterize the vaginal microbiome composition of 164 Chinese women and its association with spontaneous abortion using routine microbiological method and 16S ribosomal DNA sequencing targeting V3-V4 region.

Comments

- 1) Abstract should be rewritten and thoroughly revised. Authors should avoid the use of abbreviation. Manuscript should be revised for typos and grammatical expression. The English in the present manuscript is not of publication quality and require major improvement. Please carefully proof-read spell check to eliminate grammatical errors.
- 2) In this study, vaginal PH was measured for all the women before selecting 164 vaginal swabs for 16S rRNA sequencing. However, we do not have pertinent information on the result of vaginal Ph for these women. Briefly describe these in a table or figure.
- 3) The only exclusion criteria mentioned in the manuscript were as follows: (1) ectopic pregnancy; (2) lost follow-up; (3) took antibiotics within 30 days before sample collection; (4) had sexual activity, vaginal douching or recorded used of vaginal drugs within 48 hours before sample collection; (5) with acute inflammation, cancer of vulvovaginal candidiasis. Spontaneous Abortion can also be caused by infection. Can the authors provide information on whether the women had any history of maternal infection, history of STDs or any intercurrent disease at the course of pregnancy or previous pregnancy?

Response to reviewers

We gratefully thank the editor and all reviewers for their time spend making their constructive remarks and suggestions, which has significantly raised the quality of the manuscript and has enable us to improve the manuscript. Each suggested revision and comment were accurately considered. Our responses are given in a point-by-point manner below. In the revised version, changes to our manuscript were all highlighted within the document by using red colored text.

Reviewer 1

General Comments

Reply: Thank you for your reminding. We just finished part of the upload process in SRA and the proof is as follow. (Submission: SUB9580602 BioProject: PRJNA737055: Vaginal microbiota in early pregnancy stage).

Submission	Title	App	Status	Updated
SUB9580602	Vaginal microbiota in early pregnancy stage, May 04 '21	Sequence Read Archive (SRA)	✓ BioProject: Processed PRJNA737055 : Vaginal microbiota in early pregnancy stage BioSample: Processing (Details) Download attributes file with resulting messages	05:12

Q1: Abstract should be rewritten and thoroughly revised. Authors should avoid the use of abbreviation on the abstract without defining them initially. Manuscript should be revised for typos and grammatical expression. The English in the present manuscript is not of publication quality and require major improvement. Please carefully proof-read spell check to eliminate grammatical errors.

Reply: We feel sorry for our carelessness and poor writing. The abstract has been revised according to your suggestion marked in red. We tried our best to improve the manuscript and made some changes in the manuscript. We appreciated for your warm work earnestly, and hoped that the correction would meet with approval.

Q2: In this study, vaginal PH was measured for all the women before selecting 164 vaginal swabs for 16S rRNA sequencing. However, we do not have pertinent information on the result of vaginal PH for these women. Briefly describe this in a table or figure.

Reply: We feel sorry that we did not provide enough information about the women in our study. In the revised version, we added relevant baseline information statistics including vaginal pH about all participates, and reordered the tables in the paper accordingly. These changes will not influence the content and framework of the paper.

We hope that our additional study data will better support the findings.

Q3: The only exclusion criteria mentioned in the manuscript were as follows: (1) ectopic pregnancy; (2) lost follow-up; (3) took antibiotics within 30 days before sample collection; (4) had sexual activity, vaginal douching or recorded used of vaginal drugs within 48 hours before sample collection; (5) with acute inflammation, cancer of vulvovaginal candidiasis. Spontaneous Abortion can also be caused by infection. Can the authors provide information on whether the women had any history of maternal infection, history of STDs or any intercurrent disease at the course of pregnancy or previous pregnancy?

Reply: Thank you for your valuable suggestion. We added the history of pelvic inflammatory disease of the patients in the study, which is shown in the revised Table 1. The result indicated there was no statistically difference in the percentage of history of pelvic inflammatory disease between NP and SA group. Furthermore, we also included the history of inflammatory disease as a covariate in the analysis of Model 2. The results showed that the association between relative abundance of *Atopobium* and SA was still robust and stable.

Reviewer 2

General Comments

Reply: Thank you for your reminding. We have finished part of the upload process in SRA and the proof is as follow. (Submission: SUB9580602 BioProject: PRJNA737055: Vaginal microbiota in early pregnancy stage).

Submission	Title	App	Status	Updated
SUB9580602	Vaginal microbiota in early pregnancy stage, May 04 '21	Sequence Read Archive (SRA)	✓ BioProject: Processed PRJNA737055: Vaginal microbiota in early pregnancy stage BioSample: Processing (Details) Download attributes file with resulting messages	05:12

Q1: I suggest that the quality of this paper as currently written is not good enough for publication. It is required for English language polishing; you can have your paper professionally edited for English language. For the first time, the full name of NP and SA should be shown in the "Abstract" section.

Reply: Thank you so much for your careful check. We have rewritten the abstract and the first time a professional term appears in its full name, with abbreviations. The changes of the abstract were also highlighted by using red colored text.

Q2: Through 16S rRNA gene sequencing, the authors demonstrated that Apotobium

in the vaginal microbiota is associated with SA. However, we could not rule out the influence of pathogenic fungi such as *Candida* on SA, which might play important roles in the occurrence of SA.

Reply: Thanks for your nice suggestions. We noticed that the human vagina is a dynamic ecosystem and the vaginal ecosystem can be thrown off balance by a variety of factors. In our study, we focused on finding a new biomarker associated with the incidence of SA at the bacterial level. From the results of this study, it can be seen that at the genus level, the relative abundance of *Atopobium* was statistically different between NP and SA group. Moreover, through LEFSE analysis, relative abundance comparison and machine learning analysis, the contribution of *Atopobium* in SA group was the most significant.

Moreover, after your kind suggestion, we realized it was also important to take *Candida* and other fungi into consideration in future related study. We added this opinion in DISCUSSION in the revised manuscript marked in red.

Q3: The authors should supplement the clinicopathologic data such as histopathological analysis and analyze the potential correlation with vaginal microbiota, which contributes to make a comprehensive understanding.

Reply: Thank you for your valuable suggestion. For pregnant women, especially normal pregnant women, collecting their tissue for histopathological analysis is very harmful to their pregnancy process, which may result in miscarriage. And it is also against ethics. Therefore, in this study, we did not collect the patient's tissue for histopathological analysis. On the other hand, we also realized that relevant clinicopathologic information may help to better understand the vaginal microbiota. Further studies that include non-invasive examination and larger sample size are needed in the future. And we added this view in the DISCUSSION in the revised manuscript marker in red.

January 14, 2022

Dr. Jie Gao
First Affiliated Hospital of Guangzhou University of Chinese Medicine
Guangzhou
China

Re: Spectrum02039-21R1 (Vaginal *Atopobium* is associated with spontaneous abortion in first trimester: a prospective cohort study in China)

Dear Dr. Jie Gao:

Your manuscript has been accepted, and I am forwarding it to the ASM Journals Department for publication. You will be notified when your proofs are ready to be viewed.

Sincerely,

Kevin R. Theis
Editor, Microbiology Spectrum
